# Dynamic Characteristics of Reinforced Soil Retaining Wall with Composite Gabion Based on Time Domain Identification Method

Xiaoguang Cai [1,2,3], Shaoqiu Zhang [1], Sihan Li [1,2,3,*], Honglu Xu [4], Xin Huang [1,2,3,4], Chen Zhu [5] and Xin Liu [6]

1   College of Geological Engineering, Institute of Disaster Prevention, Sanhe 065201, China
2   Hebei Key Laboratory of Earthquake Disaster Prevention and Risk Assessment, Sanhe 065201, China
3   Key Laboratory of Building Collapse Mechanism and Disaster Prevention, China Earthquake Administration, Sanhe 065201, China
4   Key Laboratory of Earthquake Engineering and Engineering Vibration, Institute of Engineering Mechanics, China Earthquake Administration, Harbin 150080, China
5   School of Civil and Transportation Engineering, Hebei University of Technology, Tianjin 300401, China
6   Maccaferri (Changsha) New Branch Technology Development Co., Ltd., Changsha 410600, China
*   Correspondence: lisihan@st.cidp.edu.cn; Tel.: +86-187-3065-9598

**Abstract:** A series of shaking table tests was carried out on the dynamic performance and working mechanism of a gabion reinforced soil retaining wall under seismic load. The test results show that the panel presents the deformation mode of middle and upper bulging at the contact point between the rigid box and the retaining wall The settlement of top backfill is relatively uniform, and there is basically no differential settlement, the natural frequencies at different positions and heights inside the retaining wall are basically the same, and the natural frequencies are stable between 22.61 and 23.04 Hz below 0.8 g. The damping ratio decreases with the increase in wall height, and the damping ratio at each stage after vibration is greater than that before vibration. The seismic earth pressure is nonlinearly distributed. The measured value of the lower part of the retaining wall is smaller than that calculated by the Seed–Whitman method with an increase in peak acceleration, and the measured value of the upper part of the retaining wall is larger than the theoretical calculation results. The position of the resultant action point of seismic earth pressure is greater than 0.33 times the wall height specified by the Mononobe–Okabe method.

**Keywords:** reinforced soil retaining wall; shaking table test; time domain identification method; dynamic characteristic; seismic earth pressure

## 1. Introduction

The gabion reinforced soil retaining wall is a kind of flexible retaining structure. It is widely used in highways and municipal, airport and other fields because of its strong adaptability to foundation deformation [1], outstanding seismic performance, comprehensive wall water permeability and cost saving. It is a new structure of composite reinforcement of gabion and geogrids, and it is more economical and applicable for construction of tall retaining walls [2]. Geogrids prevent the formation of fracture surfaces and ensure overall and internal stability of the structure; the gabion mesh is considered to be a secondary reinforcement to ensure local stability at the panel. This new structure has been widely used at the India Sikkim Airport for the 71.5 m high slope engineering, in Albania for the 37 m high Rreshen-kalimash highway engineering, etc. In order to popularize the composite reinforced soil retaining wall structure in high intensity areas, its dynamic response should be mastered completely. Natural vibration frequency and seismic earth pressure are the key parameters of seismic design. The natural vibration frequency determines whether the reinforced soil retaining walls will resonate with the ground motion. For this reason, many scholars have studied the analytical formula and influencing factors of natural frequency

by theoretical analysis [3–7] and numerical simulation [8,9]. Based on the elastic foundation beam method, Xu et al. [3], Ghanbari et al. [4] and Ramezani et al. [5] proposed an analytical solution of natural frequency of reinforced soil retaining walls. Sarbishei and Fakher [6] solved the relationship of natural frequency of reinforced soil retaining walls by using the horizontal slice method and complex mass spring method. Darvishpour et al. [7] proposed an analytical solution of natural frequency of free vibration of the retaining walls using the Rayleigh method. Wu [8], Hatami and Bathurst [9] studied and discussed some structural design parameters affecting the natural frequency of reinforced soil retaining walls by numerical simulation. The natural frequency formulas proposed in some studies are shown in Table 1. It can be seen from Table 1 that the existing formulas regard the reinforced soil retaining wall as a whole and do not consider whether the natural frequency of the retaining wall is different at different heights, positions and peak accelerations.

**Table 1.** Previous research on fundamental frequency of reinforced walls.

| Author | Formula | Annotation |
|---|---|---|
| Xu et al. [3] | $f_j = \frac{1}{2\pi}\sqrt{\frac{\beta_j^4 EI + k_1 + k_2}{\rho A}}$ | $\beta_j$: Coefficients related to boundary conditions; E: Elastic modulus of wall panel; I: Inertia moment of homogeneous wall; $k_1$ Spring stiffness of backfill; $k_2$: Spring stiffness of reinforcement; $\rho$: Wall panel density; A: Wall panel section area. |
| Ghanbari et al. [4] | $f_c = \frac{1}{2\pi}\sqrt{\frac{12.362EI}{mL^4} + \frac{k}{m}}$ $f = \frac{1}{2\pi}\sqrt{\frac{\frac{5E}{300L^3}\left(1.701w_b w_t^2 + 4.457 w_t w_b^2 + 9.032 w_b^3 + 3.866 w_t^3\right) + 0.252kL}{\rho\,(0.049Lw_b + 0.203w_t\,L)}}$ | E: modulus of elasticity for concrete; I: Inertia moment of homogeneous wall; m: Unit length mass of homogeneous wall; L: Wall height; k: Winkler spring stiffness coefficient; $w_b$: Wall bottom width; $w_t$: Wall top width. |
| Ramezani et al. [5] | $f = \frac{1}{2\pi}\sqrt{\frac{41.7EW^3 + 0.5K_1 L^4 + 4K_2 L^3 + 86.4K_3 L + 1/2\sum_{i=0}^{n} K_i(Y(x_i)^2)}{\rho LW}}$ | E: Elastic modulus of wall; W: Width of retaining wall; L: Height of the retaining wall; $K_1$: Translational stiffness of backfill soil; $K_2$: Translational stiffness of foundation; $K_3$: Rotational stiffness of foundation; $K_i$: Axial stiffness of reinforcements; $x_i$: Reinforcement height; $\rho$: Wall density. |
| Sarbishei and Fakher [6] | $f = 2\pi\sqrt{\frac{\sum_{i=1}^{n}\frac{1}{2}k_i a_{imax}^2}{\sum_{i=1}^{n}\frac{1}{2}m_i a_{imax}^2}}$ $f_{11} = f_1 GF\ f_1 = \frac{V_s}{4H}$ $f_{11}^S = f_1 GF_s\ f_1 = \frac{V_s}{4H}$ $f_{11}^W = f_1 GF_W\ f_1 = \frac{V_s}{4H}$ | $k_i$: Reinforcement spring coefficient; $a_{imax}$: applying maximum horizontal acceleration to the slice; $m_i$: Soil quality of horizontal slices; $f_{11}$: fundamental frequencies; $V_s$: Shear wave velocity of backfill soil; H: Wall height; GF, $GF_s$, $GF_w$: Fundamental frequency determined by model size and Poisson's ratio. |
| Darvishpour et al. [7] | $f = \frac{1}{2\pi}\sqrt{0.025\frac{(kH^4 + 12.4D)g}{H^4 t\rho}}$ $f = \frac{2f_c f_t}{f_c + f_t},\ f_c = \sqrt{0.025\frac{(kH^4 + 12.4D)g}{H^4 t\rho}},\ f_t = \sqrt{0.025\frac{12.4Dg}{H^4 t\rho}}$ | k: Backfill soil stiffness; H: Wall height; D: Panel stiffness; g: Gravity acceleration; t: Panel thickness; $\rho$: Panel density. $f_c$: the fundamental frequency of the wall when it is in compression; $f_t$: the fundamental frequency of the wall when the soil is in tension. |
| Wu et al. [8] | $f = \frac{38}{H} + 0.4$ | H: Wall height. |
| Hatami and Bathurst [9] | $f = \frac{1}{CH}$ (Richardson and Lee method) $f_{11}^R = \frac{38.1}{H}$ | C: The coefficient depends on the shear modulus of backfill soil (0.02~0.033); H: Wall height. |

For the study of seismic earth pressure of a gabion retaining wall, some scholars analyzed the distribution law and the action point of the resultant force using the shaking table test. Zhu et al. [10] concluded that the peak dynamic earth pressure of gabion reinforced soil retaining walls is small in the middle and large at both ends along the wall height, which is opposite to the peak dynamic earth pressure of rigid retaining walls. Li [11]

considered that the seismic earth pressure is the sum of the static earth pressure and the dynamic earth pressure increment, which is consistent with the Seed–Whitman method. Due to the strong deformability of the gabion panel itself, the release and dissipation of seismic earth pressure is not only related to the deformation of the soil and the horizontal displacement of the panel, but also to the deformation of the panel itself. At present, there is insufficient research on the distribution of static earth pressure and dynamic earth pressure increments of gabion reinforced soil retaining walls before and after an earthquake.

It can be seen that there are few studies in the literature on the dynamic response of composite reinforced soil retaining walls under earthquake action. There is no consensus on the response characteristics of key design parameters such as natural frequency, damping ratio and seismic earth pressure of such reinforced-composite reinforced soil retaining wall. Based on this, through the shaking table test of composite gabion reinforced soil retaining wall, the response characteristics of natural frequency and damping ratio are studied via the time domain identification method, and the distribution law of seismic earth pressure and the position of resultant action point are analyzed. The results of the study will provide theoretical support for the popularization and application of the composite gabion reinforced earth retaining wall in high intensity zones.

## 2. Test Scheme

### 2.1. Shaking Table Facility

The tests were carried out on an indoor two-way electro-hydraulic servo shaking table at the China Earthquake Administration's Key Laboratory of Building Damage Mechanisms and Defense at the Institute of Disaster Prevention. Figure 1 shows the shaking table system used in the study, including a 3.0 × 1.5 × 2.3 m (length × wide × height) rigid model box. The size of the shaking table is 3.0 (length) × 3.0 m (wide), the maximum acceleration is 2.0 g, and the maximum load is 20 t (ton) [12].

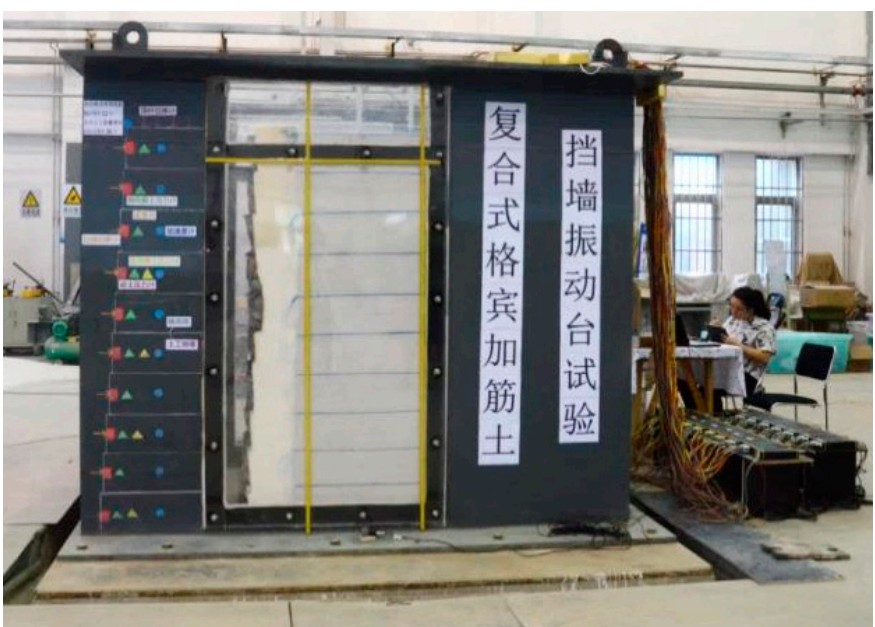

**Figure 1.** Shaking table facility with a rigid box (Shaking table test of composite gabion reinforced soil retaining wall).

### 2.2. Similitude Relationship

The commonly used height of the gabion cage in practical engineering is 0.5 or 1.0 m. Considering the bearing capacity of the shaking table and the size of the model box, the similarity constants of the composite reinforced soil retaining wall are defined as 1:5 and 1:2.5. Since the similarity relationship of all parameters cannot be satisfied in the test [13–15],

the main influence parameters are focused on according to the purpose of the experiment, and the secondary parameters are ignored. According to the similarity criterion proposed by Iai [13], the main similarity parameters of the model test are derived, as shown in Table 2.

**Table 2.** Scaling factors in the model test.

| Case Number | Parameter | Scale Factor | Scale Factor Used in This Study (Model/Prototype) | |
|---|---|---|---|---|
| | | | **1:2.5** | **1:5** |
| 1 | Length ($L$) | $C_L = 1$ | 2.5 | 5 |
| 2 | Elastic modulus ($E$) | $C_E = 1$ | 1 | 1 |
| 3 | Density ($\rho$) | $C_\rho = 1$ | 1 | 1 |
| 4 | Stress ($\sigma$) | $C_\sigma = C_E = 1$ | 1 | 1 |
| 5 | Time ($t$) | $C_t = C_L^{0.5}$ | 1.581 | 2.236 |
| 6 | Velocity ($v$) | $C_v = C_L^{0.5}$ | 1.581 | 2.236 |
| 7 | Acceleration ($a$) | $C_a = 1$ | 1 | 1 |
| 8 | Gravity ($g$) | $C_g = 1$ | 1 | 1 |
| 9 | Frequency ($\omega$) | $C_\omega = C_L^{-0.5}$ | 0.632 | 0.447 |

### 2.3. Backfill Soil

The gradation curve for badly graded medium sand for backfill ($D_{10}$ = 0.18 mm, $D_{30}$ = 0.29 mm, $D_{60}$ = 0.37 mm, Gs = 2.86, Cu = 2.06, Cc = 1.26) is shown in Figure 2. The maximum dry density is 1.99 g /cm$^3$, the minimum dry density is 1.52 g/cm$^3$, and the backfill density is 1.82 g/cm$^3$ when the relative density is 0.7. For the specific 9.3% moisture content of the triaxial unconsolidated undrained test and consolidated undrained test, use the following steps to sample: (1) Dry the sand in the oven for 6~8 h. (2) According to the moisture content of water added to the standard sand sample, stir evenly. (3) Attach the cling film into the drying cylinder, stand for 24 h, take out, and measure the actual moisture content of the sample. The actual moisture content is consistent with the design moisture content of 9.3% using the layered compaction method to quickly prepare the triaxial samples. The two test results show that the friction angles of the backfill soil are 41° and 37°, respectively. In order to strictly control the relative density, the backfill is compacted by layered filling. The specific calculation method is as follows: (1) For backfill soil drying, through the maximum minimum dry density and relative density, each layer of filling height determines the weight of each layer of soil. (2) The compaction tool is used to compact the soil layer by layer to meet the corresponding layer height. The method is consistent with that of Wang et al. [16].

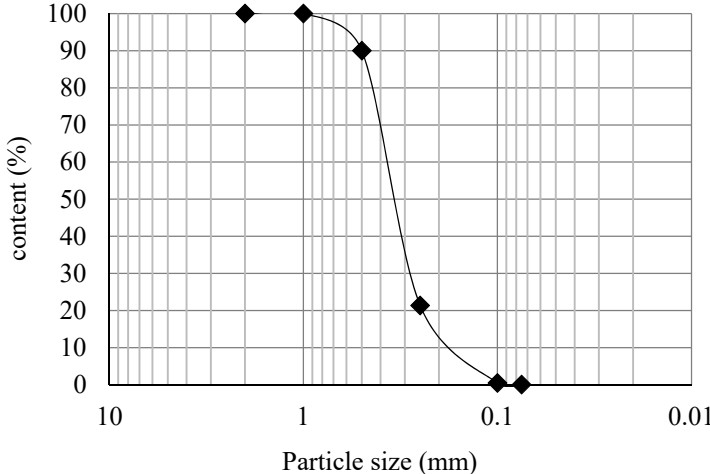

**Figure 2.** Grain size distribution curve.

### 2.4. Reinforcement Materials

The reinforced material adopts a unidirectional geogrid and gabion mesh composite structure. The length of the reinforcement is 1.4 m, the gabion mesh layer spacing is 0.2 m, and the geogrid is arranged in even layers with a layer spacing of 0.4 m. Geogrid: the length of stretching unit is 22.5 cm; the transverse rib width is 2.22 cm. The thickness is 0.1 cm; according to the similarity ratio of 1:2.5, the tensile strength of the geogrid used in the test is T2% = 6.67 kN/m and T5% = 12.23 kN/m by removing 2/3 the number of tensile elements from the geogrid. Gabion mesh: material is galvanized double-twisted hexagonal steel wire mesh; mesh diameter is 2.0 mm; mesh size is 6 × 8 cm; edge wire diameter is 3.0 mm; since the grid is a double-twisted structure, the mesh cannot be removed to reduce the strength; thus, the tensile strength of the grid is T2% = 15.8 kN/m and T5% = 19.9 kN/m.

### 2.5. Panel

The panel is made of gabion cages with the same material as the gabion mesh reinforcement. The sizes of the gabion cages are 0.75 × 0.20 × 0.20 m (length × width × height) and 0.50 × 0.20 × 0.20 m (length × width × height), as shown in Figure 3a,b, because the height of the stone cage is 0.2 m, according to the similarity ratio of 1:2.5 and 1:5, corresponding to 0.5 and 1 m, respectively. The interior is filled with hard river pebbles that are not easily weathered and that are not easily hydrolyzed. Because the mesh is too large, the back of the backfill soil can easily to lead to leakage such that in each layer after the board is laid, there is a 22 cm high geotextile, as shown in Figure 3c. In order to reduce the boundary effect between the model box and the retaining wall, a 50 mm thick sponge is added to the rear of the vibration direction to reduce the reflection of the ground motion. Vaseline is smeared on both sides of the rigid model box at the organic glass and steel plate to reduce the model restriction effect of the rigid model box on the retaining wall.

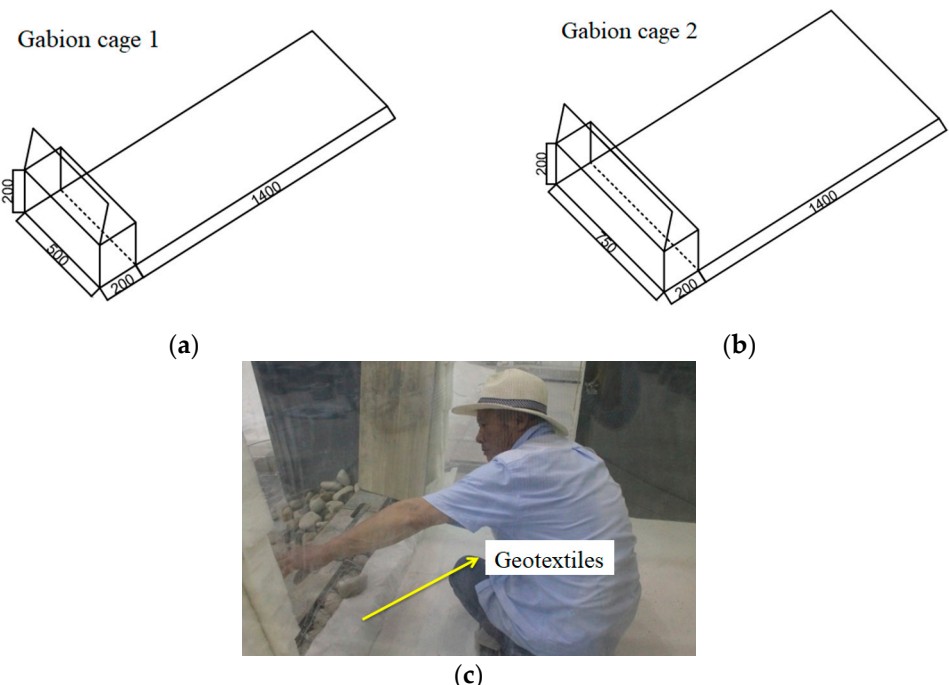

**Figure 3.** Retaining wall construction. (**a**) Gabion cage 1; (**b**) Gabion cage 2; (**c**) Geotextiles.

### 2.6. Instrument Layout

The overall size of the model is 1.9 × 1.5 × 2.0 m (length × width × height). According to the actual engineering structure, the standoff distance is set for each layer. Generally, the width of the standoff is 0.1~0.15 times the wall thickness; thus, the width of each layer is 0.02 m. Figure 4 shows the instrumentation layout of the test model. To measure the

dynamic characteristics and earth pressure response of the model, 12 accelerometers and 20 earth pressure gauges were installed in the model. Among them, ten accelerometers were installed in the reinforced area, and two accelerometers were installed on top of the wall panel to collect the acceleration time range at different locations; ten dynamic earth pressure gauges (S) were used to record the incremental values of dynamic earth pressure during the vibration process, and ten horizontal static earth pressure gauges (J) were used to measure the horizontal earth pressure of the soil before and after different loading stages. In addition, two acceleration sensors were attached to the model box for recording the input acceleration.

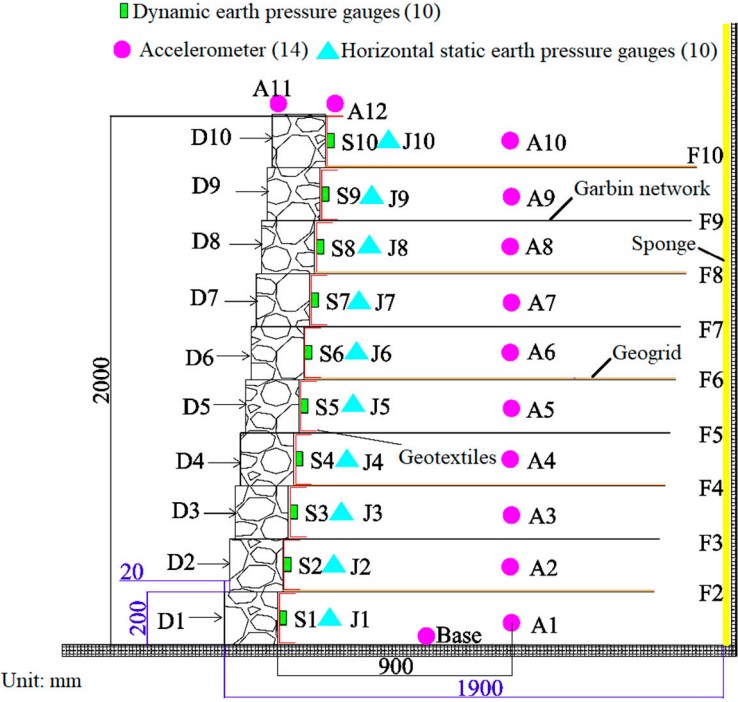

**Figure 4.** Model design.

### 2.7. Input Ground Motion

Two seismic waves were used in the model test: (1) Wolong wave (WL) recorded by Wolong station during the 2008 Wenchuan earthquake; (2) seismic waves (EL) recorded by the El-Centro Seismic Station during the 1940 Imperial Valley earthquake in the United States. The normalized one-way ground motion is input in the test. The acceleration time history after normalization and similarity processing is shown in Figure 5.

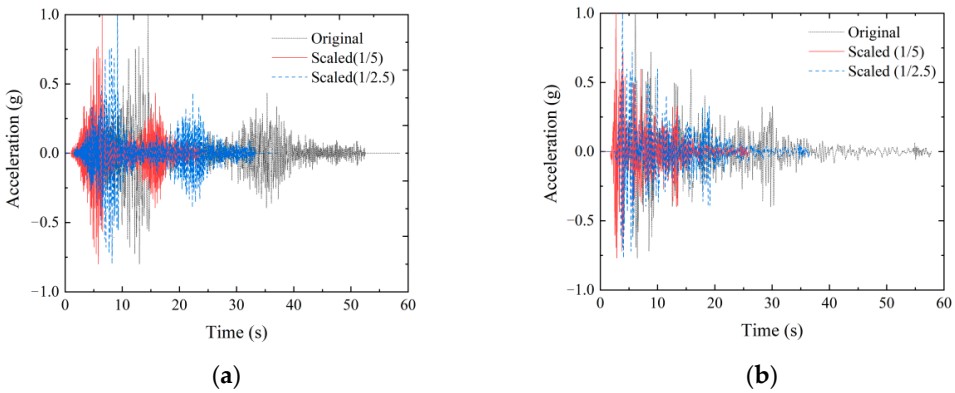

**Figure 5.** Input motions. (**a**) WL wave acceleration time history; (**b**) EL wave acceleration time history.

In order to obtain the dynamic characteristics of the model, white noise (WN) is input before and after each acceleration amplitude change. Although continuous loading changes the initial state of the model, more information can be obtained from the model, similar to the type of seismic loading used in many shaking table tests [14,15,17]. The loading conditions are shown in Table 3.

**Table 3.** Loading cases.

| Case Number | Waveform | Amplitude/g | Similarity Ratio | Condition Code |
|---|---|---|---|---|
| | WN | 0.05 | 1 | WN1 |
| 1, 2 | WL, EL | 0.1 | 5 | WL0.1 g, EL0.1 g |
| 3, 4 | WL, EL | 0.1 | 2.5 | WL0.1 g, EL0.1 g |
| | WN | 0.05 | 1 | WN2 |
| 5, 6 | WL, EL | 0.2 | 5 | WL0.2 g, EL0.2 g |
| 7, 8 | WL, EL | 0.2 | 2.5 | WL0.2 g, EL0.2 g |
| | WN | 0.05 | 1 | WN3 |
| 9, 10 | WL, EL | 0.4 | 5 | WL0.4 g, EL0.4 g |
| 11, 12 | WL, EL | 0.4 | 2.5 | WL0.4 g, EL0.4 g |
| | WN | 0.05 | 1 | WN4 |
| 13, 14 | WL, EL | 0.6 | 5 | WL0.6 g, EL0.6 g |
| 15, 16 | WL, EL | 0.6 | 2.5 | WL0.6 g, EL0.6 g |
| | WN | 0.05 | 1 | WN5 |
| 17, 18 | WL, EL | 0.8 | 5 | WL0.8 g, EL0.8 g |
| 19, 20 | WL, EL | 0.8 | 2.5 | WL0.8 g, EL0.8 g |
| | WN | 0.05 | 1 | WN6 |
| 21, 22 | WL, EL | 1.2 | 5 | WL1.2 g, EL1.2 g |
| 23, 24 | WL, EL | 1.2 | 2.5 | WL1.2 g, EL1.2 g |
| | WN | 0.05 | 1 | WN7 |
| 25 | WL | 1.6 | 5 | WL1.6 g |
| 26 | WL | 1.6 | 2.5 | WL1.6 g |
| | WN | 0.05 | 1 | WN8 |
| 27 | WL | 2.0 | 5 | WL2.0 g |
| 28 | WL | 2.0 | 2.5 | WL2.0 g |
| | WN | 0.05 | 1 | WN9 |

## 3. Test Result

### 3.1. Experimental Phenomena

Due to the terrain conditions, engineering purposes and other factors, the gabion-type reinforced soil retaining wall project at some positions is connected with the tunnel project. This situation is very common in China's highway and railway projects, such as the tunnel entrance of Yichang Expressway in Jiangsu Province shown in Figure 6 [18]. The structure and the overall stiffness of the two structures are different, resulting in the displacement distribution of the reinforced soil retaining wall panel and the settlement distribution of backfill being different at the stiffness mutation and inside the retaining wall.

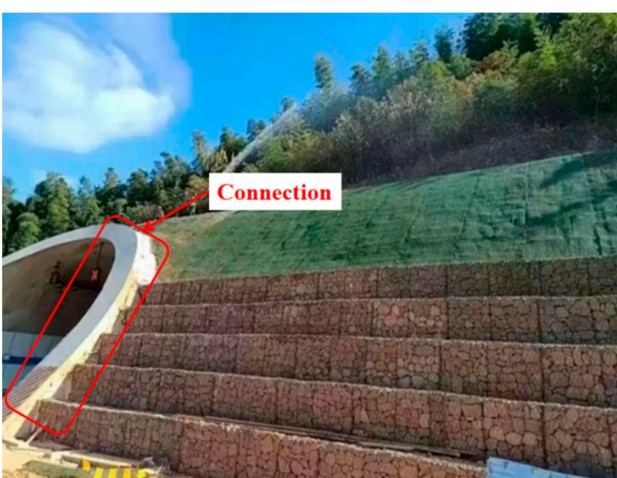

**Figure 6.** Jiangsu Section Tunnel Entrance of Yichang Expressway.

In order to analyze the panel displacement and backfill settlement law of the reinforced earth retaining wall at the stiffness mutation, 0.5 cm thick blue sand was laid every 20 cm vertically to record the backfill settlement law at different heights. At the same time, the locations of the panels and backfill were recorded on the organic glass before the start of the test and after each working condition, as shown in Figure 7.

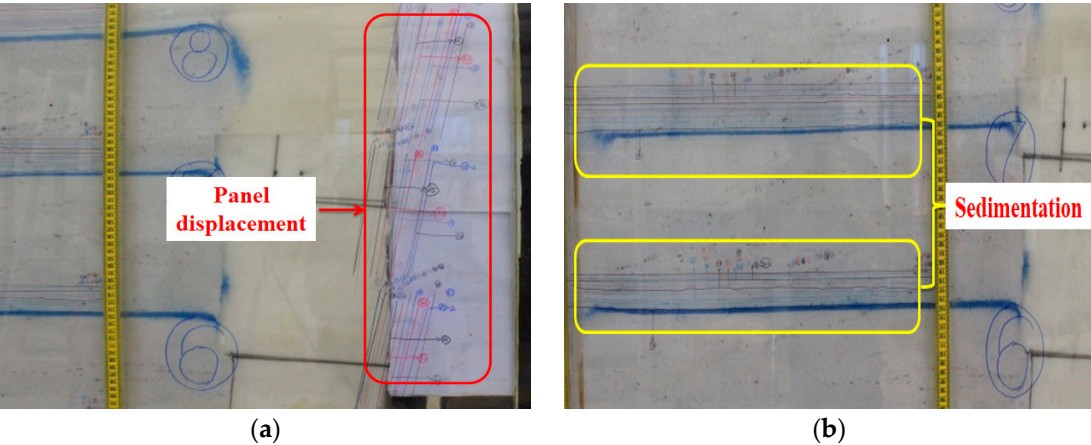

**(a)**                                                                     **(b)**

**Figure 7.** Macroscopic records at variable stiffness. (**a**) Panel displacement records; (**b**) backfill settlement records.

From Figure 8a, it can be seen that the deformation pattern of the retaining wall is bulging in the middle and upper part, but the bulging part moves upward with the increase in seismic load, while the top displacement grows obviously and is only slightly smaller than the horizontal displacement values of the D7 and D8 layers. In the working condition of 1–20, the maximum deformation of the retaining wall appears in the D7 layer (130 cm); under the condition of 21–28, the bulging position is transferred from the D7 layer to the D8 layer (150 cm), and the horizontal displacement at the top of the panel is slightly smaller than the maximum bulging deformation. From Figure 8b, it can be seen that the overall settlement of the gabion-type retaining wall is more uniform, and there is basically no differential settlement. In working conditions 1–16 (i.e., the peak acceleration does not exceed 0.6 g), the settlement change value is small and the increase rate is slow; in working conditions 17–28, the increase rate of the settlement value is significantly accelerated, and the increase rate is faster when the acceleration amplitude or similar constants are changed, while the settlement increase in the two earthquake loads is smaller when the acceleration amplitude and similar constants are the same.

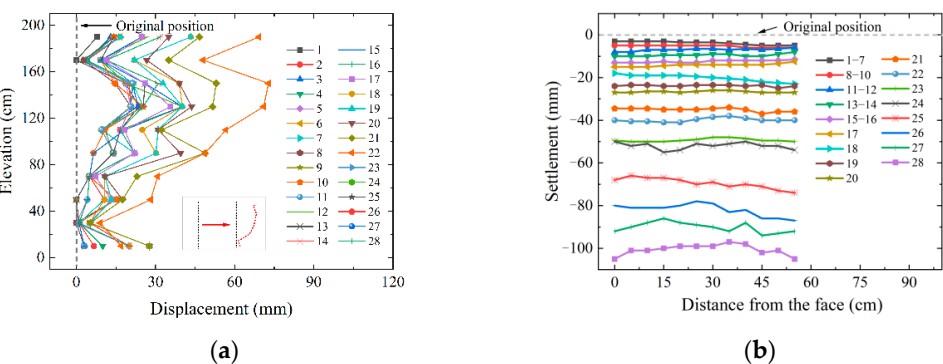

**(a)**                                                                     **(b)**

**Figure 8.** The field observation of the gabion retaining wall. (**a**) Wall deformation observation; (**b**) top blue sand settlement.

*3.2. Dynamic Behavior*

The dynamic characteristics of the reinforced soil retaining wall mainly include frequency, damping and vibration mode [19,20]. The purpose of inputting white noise in the model test is to obtain its dynamic characteristics. Judging the natural frequency of the structure is the key to avoid the resonance between the structure and the site frequency. Yazdandoust [17] used two methods to calculate the natural frequency (F) of the reinforced soil retaining wall: (1) using the simplified formula: $F = \frac{V_s}{4H}$, where '$V_s$' is the shear wave velocity of the reinforced soil structure, 'H' is the wall height; (2) spectrum analysis of free vibration response after using shock pulse.

There are two kinds of calculation theories for solving dynamic characteristics: frequency domain identification method and time domain identification method. (1) The frequency domain method is mainly reflected by the frequency response function (also known as the transfer function). The frequency response function is the quotient of the cross power spectral density function and the self-power spectral density function. The self/cross power spectral density function is converted from the self/cross correlation function by the Welch method (also known as the average periodogram method). (2) The time domain method is used to preprocess the input data by the random decrement method, and then identifies the dynamic characteristics by the least square method, ITD method and STD method. In this section, the time domain identification method (STD method) is used to calculate the natural frequency and damping ratio.

The first-order natural frequency and damping ratios of the gabion retaining wall under different working conditions are shown in Table 4. The results show that: (1) The natural frequency of the top gabion cage (A11) is basically consistent with the natural frequency of the backfill soil (A1, A2, A3, A4, A5, A6, A7, A8, A9, A10) and the vibration table (Base), which is consistent with the test results of Wei Ming et al. [20]. Comparing different loading stages, the natural frequency value is 22.27~24.48 Hz, which is consistent. (2) With the proceeding of loading, the position of maximum damping ratio decreases gradually from A4 (before vibration) to A3 (After 0.1 g) and then to A1 (After 0.2 g). The damping ratio decreases with the increase in wall height. The reason for this phenomenon is that the shear strain of the soil increases with the increase in buried depth [21] (See Equations (1) and (2)).

$$\tau(z) = \frac{\pi G}{4H} u_{max} \sin \frac{\pi z}{2H} = G\gamma \tag{1}$$

$$so, \gamma = \frac{\pi}{4H} u_{max} \sin \frac{\pi z}{2H} \tag{2}$$

Here, $\tau$ (z) is the shear stress at depth z; G is the dynamic shear modulus; $u_{max}$ is the maximum horizontal displacement of ground surface; z is the depth of soil from the surface; $\gamma$ is shear strain. It can be seen that the shear strain is a sine function changing with depth. The greater the shear strain, the more prone to plastic deformation ($\sigma = E\varepsilon = c\dot{u}$, c is damping). Large plastic deformation leads to strong hysteretic energy dissipation, further indicating large damping. (3) The damping ratio after vibration is slightly larger than that before vibration, which indicates that the vibration leads to the increase in shear strain in the reinforced area, and the soil is further compacted.

**Table 4.** Dynamic characteristics of gabion retaining wall.

| Position | Before Vibration | | After 0.1 g | | After 0.2 g | | After 0.4 g | |
|---|---|---|---|---|---|---|---|---|
| | Frequency (Hz) | Damping Ratio (%) | Frequency (Hz) | Damping Ratio (%) | Frequency (Hz) | Damping Ratio (%) | Frequency (Hz) | Damping Ratio (%) |
| Base | 22.91 | 2.73 | 23.46 | 4.24 | 23.14 | 4.31 | 23.16 | 3.84 |
| A1 | 22.85 | 3.96 | 24.48 | 9.59 | 23.89 | 7.29 | 23.35 | 7.43 |
| A2 | 22.93 | 3.09 | 24.04 | 12.43 | 23.57 | 6.65 | 23.08 | 6.75 |
| A3 | 22.94 | 3.83 | 23.55 | 14.29 | 23.05 | 6.71 | 22.86 | 7.83 |
| A4 | 22.94 | 4.36 | 22.81 | 7.86 | 22.82 | 6.16 | 23.00 | 6.42 |
| A5 | 22.71 | 4.21 | 22.48 | 5.98 | 22.45 | 4.63 | 22.65 | 6.03 |
| A6 | 22.44 | 4.30 | 22.17 | 4.51 | 22.29 | 3.40 | 22.28 | 5.80 |
| A7 | 22.36 | 3.17 | 22.55 | 3.73 | 22.39 | 3.21 | 22.27 | 3.82 |
| A8 | 22.44 | 3.09 | 22.66 | 3.45 | 22.50 | 3.24 | 22.37 | 3.58 |
| A9 | 22.43 | 3.03 | 22.71 | 3.09 | 22.54 | 3.15 | 22.38 | 3.27 |
| A10 | 22.45 | 2.68 | 22.78 | 3.09 | 22.55 | 3.18 | 22.37 | 3.09 |
| A11 | 22.80 | 2.17 | 23.17 | 3.54 | 22.85 | 2.77 | 22.74 | 3.02 |

The natural frequency and damping ratio of each measuring point in the gabion retaining wall structure (A1, A2, A3, A4, A5, A6, A7, A8, A9, A10, A11) are taken as the natural frequency and damping ratio of the gabion retaining wall and are compared with some methods (Wu method [15], Richardson and Lee method [9]). The distribution of natural frequency and damping ratio of the gabion retaining wall in different loading stages is shown in Figure 9. The data show that: (1) The basic frequency of the natural frequency is stable at 22.61~23.04 Hz in different vibration stages, and there is no significant change. The overall amplitude is within the prediction interval of the Richardson and Lee methods, and it is close to the calculated value of the Wu method. (2) The change trend of the damping ratio is nonlinear, ranging from 3.45% to 6.50%. The damping ratio of each stage after vibration is greater than that before vibration.

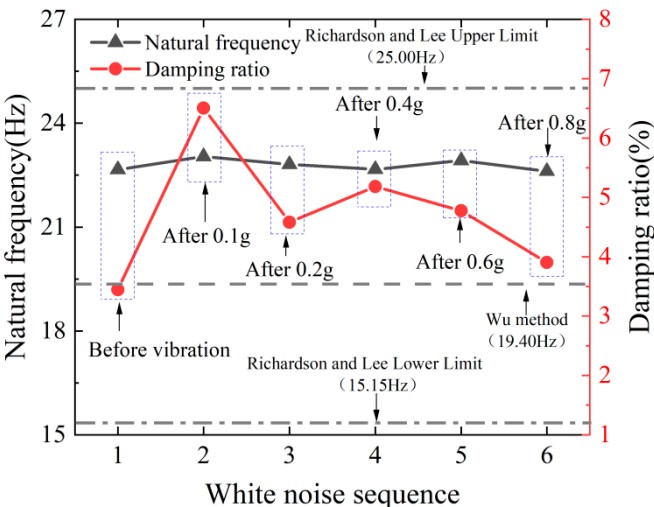

**Figure 9.** Distribution law of natural frequency and damping ratio.

### 3.3. Dynamic Behavior

One side of the reinforced soil retaining wall is backfill area, and the other side is free face. When the wall moves to the free face, the soil produces active earth pressure; when the wall moves to the backfill area, passive earth pressure is generated. It is generally believed that the passive earth pressure will not cause damage to the retaining wall; thus, the directionality of the earth pressure should be paid attention to when processing the test data. This section analyzes the active earth pressure generated by the soil. The Mononobe–Okabe (M-O) method [22] and the Seed–Whitman (S-W) method [23] are the two main numerical methods. The M-O method regards the seismic force as a quasi-static force acting on the center of gravity of the sliding wedge on the back of the wall and is derived from

the static equilibrium condition of the sliding wedge according to the Coulomb theory. The resultant force is applied at the height of the H/3 wall. The S-W method considers that the seismic earth pressure is the resultant force of the static earth pressure and the seismic dynamic earth pressure increment, and the resultant force points are H/3 (static earth pressure) and 3H/5 (dynamic earth pressure increment), respectively. Studies have shown that the distribution of earth pressure is not a linear distribution but a curve distribution.

Through the measured static earth pressure and dynamic earth pressure increment data to analyze the static earth pressure distribution and resultant force action point, dynamic earth pressure distribution and resultant force action point and seismic active earth pressure distribution and resultant force action point, the influence of seismic load on earth pressure distribution is explored, as shown in Figures 10–15. The measured results are represented by M; the results of the theoretical calculation (S-W method) are represented by C; the title of the working condition in the horizontal earth pressure is the measured value (R) after the vibration of the working condition; the seismic active earth pressure is the sum of the measured static earth pressure and the dynamic earth pressure increment. It can be seen from Figure 10 that the overall law of horizontal static earth pressure only changes slightly before and after multiple vibrations after the completion of construction: the middle and lower parts (F5 layer and below) increase slightly with the gradual increase in vibration; the upper part decreases slightly with the increase in vibration. This is due to the multiple vibrations in the middle and lower parts of the increasingly dense soil. The top is due to the weak constraint, which makes the displacement increase, caused by the release of earth pressure. Compared with the theoretical method, the measured value is greater than the calculation results of the active earth pressure and static earth pressure. Although the distribution along the wall height is nonlinear, the overall trend is consistent with the static earth pressure. The static earth pressure resultant force action point (as shown in Figure 11) position with multiple vibrations, from 0.41 H after the completion of the construction, gradually reduced to 0.38 H, which is due to the overall tilt of the retaining wall. Meanwhile, the measured value is slightly larger than 1H/3 of the linear distribution of the earth pressure.

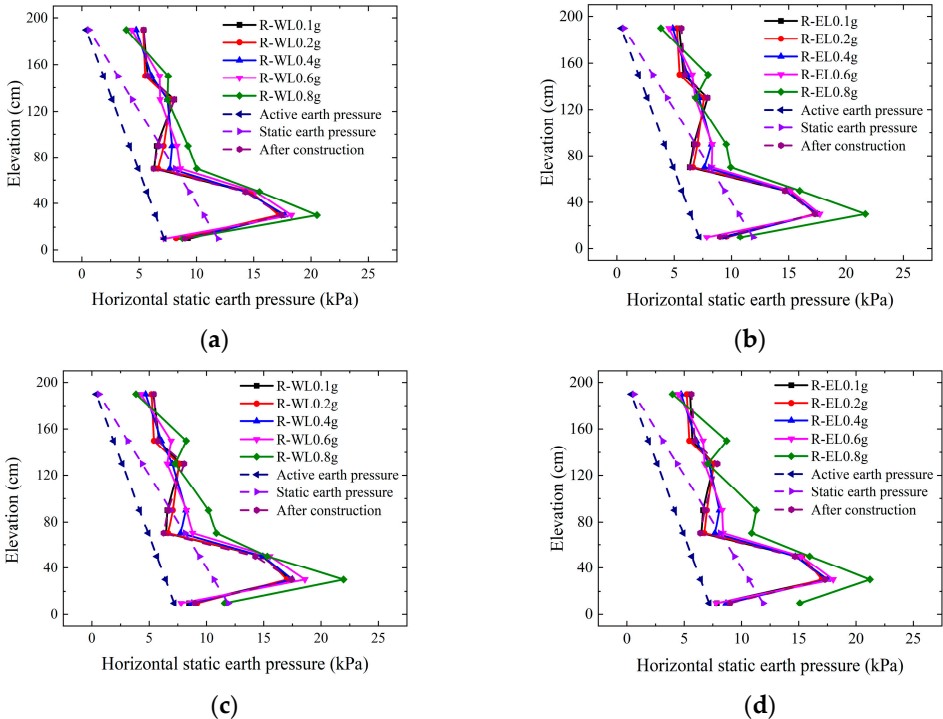

**Figure 10.** Distribution law of lateral static earth pressure. (**a**) WL wave (scale factor:1:5); (**b**) EL wave (scale factor:1:5); (**c**) WL wave (scale factor:1:2.5); (**d**) EL wave (scale factor:1:2.5).

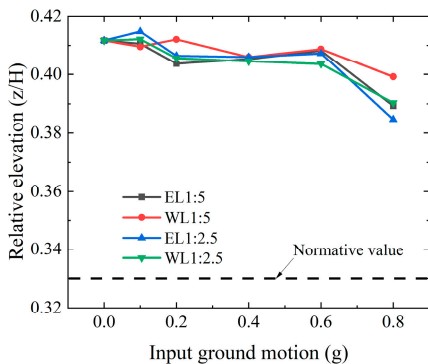

**Figure 11.** Position of action point of lateral static earth pressure resultant force.

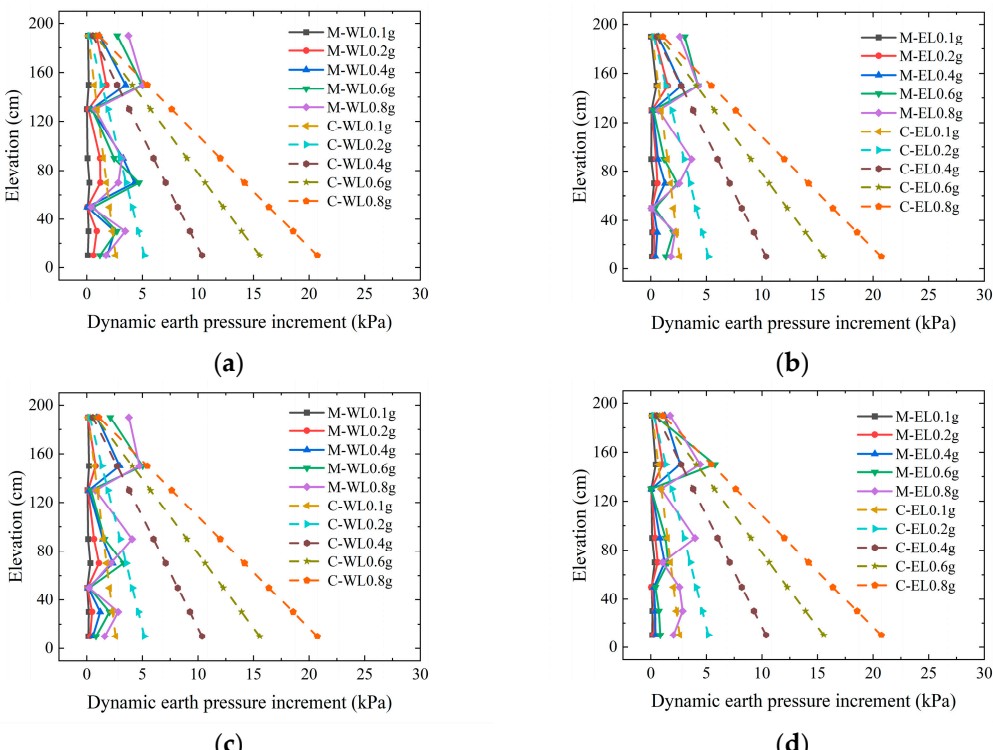

**Figure 12.** Distribution law of dynamic earth pressure increment. (**a**) WL wave (scale factor:1:5); (**b**) EL wave (scale factor:1:5); (**c**) WL wave (scale factor:1:2.5); (**d**) EL wave (scale factor:1:2.5).

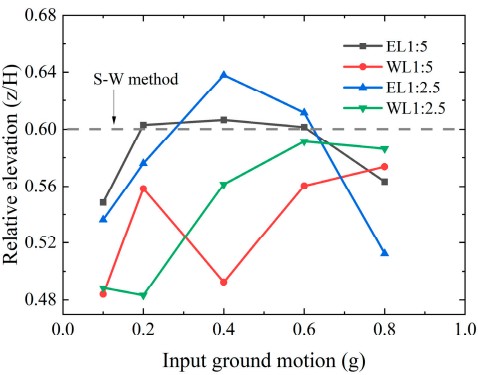

**Figure 13.** Position of resultant force action point of dynamic earth pressure increment.

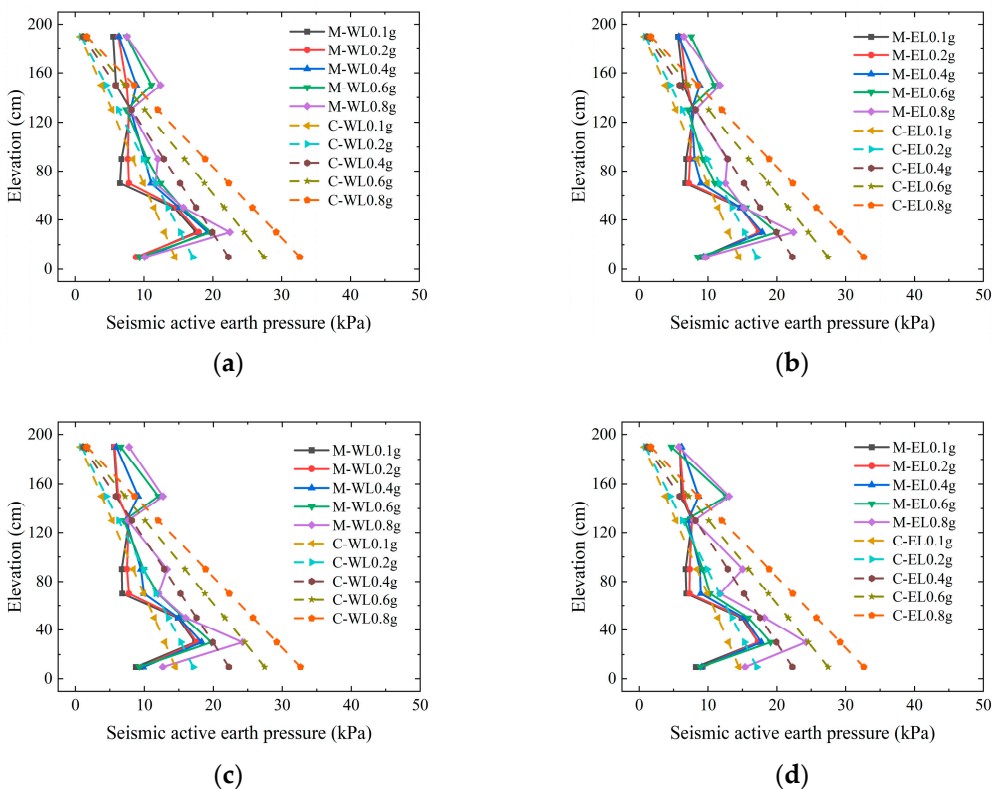

**Figure 14.** Distribution law of seismic active earth pressure. (**a**) WL wave (scale factor:1:5); (**b**) EL wave (scale factor:1:5); (**c**) WL wave (scale factor:1:2.5); (**d**) EL wave (scale factor:1:2.5).

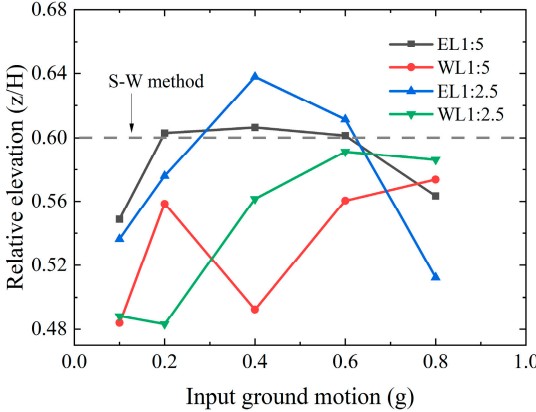

**Figure 15.** Location of resultant force acting point of seismic active earth pressure.

The distribution of the dynamic earth pressure increment of the gabion retaining wall (see Figure 12) is less affected by the peak acceleration, and it only changes greatly at the top of the retaining wall. The main reason is the influence of the panel stiffness: the lateral deformation of the gabion panel itself can offset some of the dynamic earth pressure increment, which is consistent with the conclusion of Zhu [10]. Compared with the theoretical value, the measured value is less than the theoretical value, and the greater the peak acceleration, the greater the gap. The reason for this phenomenon is that the calculation of the theoretical value is assumed to be based on a rigid retaining wall based on a semi-infinite elastomer. The position of the action point of the resultant force of the dynamic earth pressure increment (see Figure 13) is in the range of 0.48 H~0.64 H. Compared with the 0.6 H position in the S-W method, when the EL wave is 0.2~0.6 g, the position of the resultant action point is basically consistent with the standard value, and

the rest values are less than the standard value. In addition, the resultant force point of the WL wave is generally lower than that of the EL wave, which is related to the energy and spectral characteristics of the two seismic waves.

The distribution trend of seismic active earth pressure is shown in Figure 14. From Figure 13, it can be seen that the increment of dynamic earth pressure in the middle and lower regions is small; thus, the distribution trend of seismic active earth pressure in the middle and lower regions is consistent with the distribution law of measured static earth pressure. With the increase in peak acceleration, the overall trend of the lower measured value gradually changes from distributing on both sides of the standard value (0.1~0.2 g) to less than the standard value (0.4~0.8 g). The upper part of the retaining wall is affected by static earth pressure and dynamic earth pressure increment, and the theoretical value is always less than the measured value. The position of the resultant action point of seismic active earth pressure (see Figure 15) shows an overall trend of 'rising first and then falling'. This tendency is consistent with Shin's rule [24], which is opposite to the trend of gradual decrease in static earth pressure. From the range of resultant force action point (0.41~0.44 H), it can be seen that the position of the gabion wall action point is mainly affected by the distribution of static earth pressure when it is 0.1~0.4 g, and by the dynamic earth pressure increment and static earth pressure under strong earthquakes (0.6 and 0.8 g).

## 4. Discussion

When designing the model test of the reinforced soil retaining wall, in order to accurately measure the test results and return the model to the prototype analysis, the similarity between the prototype and the test model should be considered. The test results of the scale test depend on whether the mechanical properties of each part of the material are reasonable. In this test, the size of the geogrid and model meet the requirements of the similarity ratio, but it is difficult to find the materials that match the mechanical properties of the pebble and backfill in the gabion cage. Therefore, the pebbles and backfill sand in this test are not scaled, which is an inevitable problem in the model shaking table test.

Therefore, in the later stage, it is still necessary to carry out on-site testing and numerical simulations to further verify the research results of this paper, which will help to understand the seismic performance of the gabion reinforced soil retaining wall more accurately and comprehensively, so as to provide information for the subsequent seismic design of the reinforced soil retaining wall.

This paper is only a single-model multi-condition shaking table test and does not further compare the test results through on-site monitoring and numerical simulation modeling. It will continue to improve and supplement in follow-up work.

## 5. Conclusions

At the stiffness mutation position (i.e., the contact between the rigid box and the reinforced soil retaining wall), the wall shows a deformation mode of bulging in the middle and upper parts: as the test condition progresses, the maximum displacement of the bulging deformation is transferred from the D7 layer to the D8 layer; the displacement at the top increases obviously, which is only slightly smaller than the horizontal displacement value of D7 and D8 layers. The top settlement is relatively uniform as a whole, and there is basically no differential settlement. With the increase in peak acceleration, the growth rate of the settlement value increases.

The natural frequencies at different positions and heights are basically the same, ranging from 22.27 to 24.48 Hz. The natural frequency of the whole retaining wall is stable between 22.61 and 23.04 Hz in different vibration stages (below 0.8 g). The damping ratios at different heights are different and decrease with the increase in wall height. The overall trend is nonlinear, ranging from 3.45% to 6.50%, and the damping ratios at each stage after vibration are greater than those before vibration.

The conclusion of earth pressure analysis has three parts: (1). The measured value of horizontal static earth pressure is greater than the theoretical calculation value (active earth

pressure, static earth pressure), and the overall distribution is nonlinear; with the development of the vibration condition, the resultant point of the static earth pressure decreases from 0.41 to 0.38 H, which is greater than the theoretical value of 0.33 H. (2) The distribution of dynamic earth pressure increment is non-linear and is less affected by peak acceleration, and the measured results are less than those calculated by the S-W method; The position of the resultant force is between 0.48 and 0.64 H and fluctuates at 0.6 H (S-W method). (3) With the increase in peak acceleration, the overall trend of seismic active earth pressure in the middle and lower parts of the wall is gradually smaller than that of the standard value (0.1~0.2 g) (0.4~0.8 g). The measured value of the upper retaining wall is always greater than the theoretical value; the position of the resultant force of seismic active earth pressure is 0.41~0.44 H, which is greater than 0.33 H (M-O method).

**Author Contributions:** Methodology, X.C.; conceptualization, S.Z.; writing—original draft preparation, S.L.; data curation, H.X.; formal analysis, C.Z.; validation, X.H.; resources, X.L. All authors have read and agreed to the published version of the manuscript.

**Funding:** This research was funded by the Earthquake Technology Spark Program of China, grant number XH204402; the Earthquake Technology Spark Program of China, grant number XH23067YA; the Fundamental Research Funds for the Central Universities, grant number ZY20215107; and the Langfang Science and Technology Support Plan Project, grant number 2021013172.

**Institutional Review Board Statement:** Not applicable.

**Informed Consent Statement:** Not applicable.

**Data Availability Statement:** Not applicable.

**Acknowledgments:** The writers appreciate Jiayu Feng and Xuepeng Wang for their help in data-assisted analysis.

**Conflicts of Interest:** The authors declare no conflict of interest.

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
