# Peer review of "Dynamic Characteristics of Reinforced Soil Retaining Wall with Composite Gabion Based on Time Domain Identification Method"

_sustainability, doi:10.3390/su142316321_

Round 1
Reviewer 1 Report
In this paper, gabion cage was used as wall panel, and the gabion and geogrid were used as the composite reinforcement materials. The test model of the composite reinforced soil retaining wall was made and the shaking table test was carried out. The wall deformation, top settlement, the distribution of natural frequency, damping ratio, the distribution of seismic earth pressure and the distribution of resultant force point were analyzed. The research purpose is clear, and the research object has strong engineering background and application potential. It is recommended to accept after modification:
1、It is suggested that the calculation formula of natural frequency in Table 1 should be arranged according to the serial number of references. And explain the fc and ft in the Darvishpour’s reference.
2、References 16 were cited in Sections 1.3 and 2.2. Please check whether the references 15 and 16 are correct.
3、In section 1.1, what is the meaning of the parameters ' L, W, H, t ', it is recommended to explain in detail.
4、Regarding the similarity relationship in section 1.2, the height of the gabion cage is generally 0.5m or 1.0m, and the similarity constants are defined as 1:5 and 1:2.5. Is it correct to say that 0.5m corresponds to 1:2.5 and 1.0m corresponds to 1:5?
5、What is the meaning and expression of Figure 3 ( c ) ? Not useful for the whole, it is recommended to delete.
6、Size units should be added in Fig.4.
7、The spacing between the layers is 20cm. Why is there a layer of blue sand 15cm in Section 2.1? In figure 8 (a), please adjust the line thickness to ensure that the displacement curve under each working condition is clear.
8、In the explanation of the damping ratio in section 2.2, the influence of shear strain on it is mentioned. Can the shear stiffness and shear strain be obtained by processing the relevant data, so as to analyze the change of damping ratio more accurately?
9、The input condition in Table 3 is WL2.0g, why only 0.8g is analyzed in Section 2.2 and Section 2.3?
10、Can the change of natural frequency and damping ratio know the damage state of retaining wall ?
11、Figure 9 shows the change of natural frequency and damping ratio with different vibration stages. Wu's calculation formula was selected for comparison. The change rule in the paper is as follows: with the increase of wall height, the first-order frequency is inversely proportional to the wall height, and the decrease is large. When the wall height of the gabion panel reinforced earth retaining wall increases, the frequency increase range is small. What is the reason?
12、In the analysis of earth pressure in section 2.3, the measured static earth pressure value is larger than the theoretical calculation value. Has the author considered the reason?
Author Response
I am very grateful to the experts for their recognition of my work, and put forward professional opinions on improving the quality of the paper, so that the quality of the manuscript has been greatly improved. The author revised the manuscript according to the expert opinion.
Please see the attachment for the specific reply.

Reviewer 2 Report
This paper, entitled Dynamic characteristics of reinforced soil retaining wall with composite gabion based on time domain identification method, is a scholarly work and can increase knowledge on this domain.
The authors provide an interesting and original study, the content is relevant to ‘Sustainability’. The abstract and keywords are meaningful. The manuscript is quite well written and well related to existing literature.
I have some general and specific comments:
1. Since the position of resultant action point of seismic earth pressure is adopted to interpret the test phenomenon following the existing literature (Shin et al. (2019), “Numerical simulation and shaking table test of geotextile bag retaining wall structure”), the following literature should be cited.
2. The model box is made of steel plate and steel bar. The model box will confine the retaining wall model greatly. How to deal with the boundary effect?
3. Different from the observation of many rigid retaining structures, the acceleration response of composite gabion retaining wall presents a decreasing trend along its height. Please give a reasonable interpretation regarding this issue.
4. How to make a difference between static earth pressure and dynamic earth pressure in shaking table test. Please give a brief explanation.
5. One of the significant concerns is that the authors should carefully develop a discussion section to talk about the significance, shortages or advantages of the methods you proposed, the reliability and meaning of your results (compared to other related studies) etc.
6. I didn’t find anything related to the limitations of the research. It is recommended to complete it.
7. What are the perspectives of this work? What are the future experiments?
8. As it, this manuscript is not fully acceptable for publication and requires some amendments and additional data or information. I recommend to revise this manuscript according to the previous comments.
Author Response

(The authors gave the same response as above.)

Round 2
Reviewer 2 Report
In the revised manuscript, the comments I pointed out were appropriately corrected.
Therefore, ‘Accept in present form’ is recommended.